# REPRESENTING FORMAL LANGUAGES:
# A COMPARISON BETWEEN FINITE AUTOMATA
# AND RECURRENT NEURAL NETWORKS

**Joshua J. Michalenko**
Rice University
jjm7@rice.edu

**Ameesh Shah**
Rice University
ars7@rice.edu

**Abhinav Verma**
Rice University
averma@rice.edu

**Richard G. Baraniuk**
Rice University
richb@rice.edu

**Swarat Chaudhuri**
Rice University
swarat@rice.edu

**Ankit B. Patel**
Baylor College of Medicine, Rice University
abp4@rice.edu

## ABSTRACT

We investigate the internal representations that a recurrent neural network (RNN) uses while learning to recognize a regular formal language. Specifically, we train a RNN on positive and negative examples from a regular language, and ask if there is a simple decoding function that maps states of this RNN to states of the minimal deterministic finite automaton (MDFA) for the language. Our experiments show that such a decoding function indeed exists, and that it maps states of the RNN not to MDFA states, but to states of an *abstraction* obtained by clustering small sets of MDFA states into "superstates". A qualitative analysis reveals that the abstraction often has a simple interpretation. Overall, the results suggest a strong structural relationship between internal representations used by RNNs and finite automata, and explain the well-known ability of RNNs to recognize formal grammatical structure.

## 1 INTRODUCTION

Recurrent neural networks (RNNs) seem "unreasonably" effective at modeling patterns in noisy real-world sequences. In particular, they seem effective at recognizing grammatical structure in sequences, as evidenced by their ability to generate structured data, such as source code (C++, LaTeX, etc.), with few syntactic grammatical errors (Karpathy et al., 2015). The ability of RNNs to recognize *formal languages* – sets of strings that possess rigorously defined grammatical structure – is less well-studied. Furthermore, there remains little systematic understanding of *how* RNNs recognize rigorous structure. We aim to explain this internal algorithm of RNNs through comparison to fundamental concepts in formal languages, namely, finite automata and regular languages.

In this paper, we propose a new way of understanding how trained RNNs represent grammatical structure, by comparing them to finite automata that solve the same language recognition task. We ask: Can the internal knowledge representations of RNNs trained to recognize formal languages be easily mapped to the states of automata-theoretic models that are traditionally used to define these same formal languages? Specifically, we investigate this question for the class of *regular languages*, or formal languages accepted by *finite automata* (FA).

In our experiments, RNNs are trained on a dataset of positive and negative examples of strings randomly generated from a given formal language. Next, we ask if there exists a *decoding function*: an isomorphism that maps the hidden states of the trained RNN to the states of a canonical FA. Since there exist infinitely many FA that accept the same language, we focus on the *minimal deterministic finite automaton* (MDFA) — the deterministic finite automaton (DFA) with the smallest possible number of states – that perfectly recognizes the language.

Our experiments, spanning 500 regular languages, suggest that such a decoding function exists and can be understood in terms of a notion of *abstraction* that is fundamental in classical system theory. An *abstraction* $\mathcal{A}$ of a machine $\mathcal{M}$ (either finite-state, like an FA, or infinite-state, like a RNN) is a machine obtained by clustering some of the states of $\mathcal{M}$ into "superstates". Intuitively, an abstraction

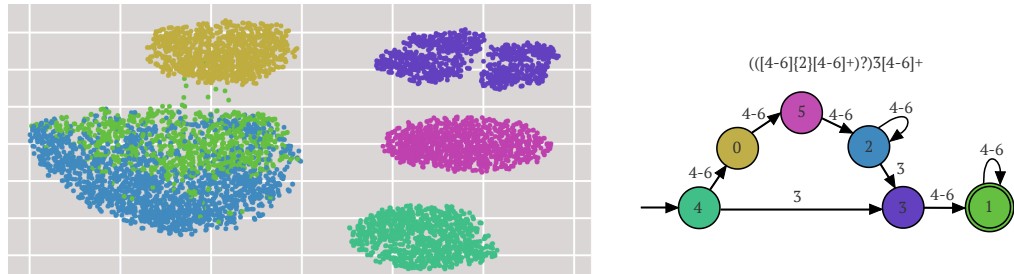

Figure 1: $t$-SNE plot (Left) of the hidden states of a RNN trained to recognize a regular language specified by a 6-state DFA (Right). Color denotes DFA state. The trained RNN has abstracted DFA states 1(green) and 2(blue) (each independently model the pattern `[4-6]*`) into a single state.

$\mathcal{A}$ loses some of the discerning power of the original machine $\mathcal{M}$, and as such recognizes a superset of the language that $\mathcal{M}$ recognizes. We observe that the states of a RNN $\mathcal{R}$, trained to recognize a regular language $\mathcal{L}$, commonly exibit this abstraction behavior in practice. These states can be decoded into states of an abstraction $\mathcal{A}$ of the MDFA for the language, such that with high probability, $\mathcal{A}$ accepts any input string that is accepted by $\mathcal{R}$. Figure 1 shows a t-SNE embedding (Maaten and Hinton, 2008) of RNN states trained to perform language recognition on strings from the regex `[(([4-6]{2}[4-6]+)?)3[4-6]+]`. Although the MDFA has 6 states, we observe the RNN abstracting two states into one. Remarkably, a *linear* decoding function suffices to achieve maximal decoding accuracy: allowing nonlinearity in the decoder does not lead to significant gain. Also, we find the abstraction has low "coarseness", in the sense that only a few of the MDFA states need be clustered, and a qualitative analysis reveals that the abstractions often have simple interpretations.

## 2    RELATED WORK

RNNs have long been known to be excellent at recognizing patterns in text (Kombrink et al., 2011; Karpathy et al., 2015). Extensive work has been done on exploring the expressive power of RNNs. For example, finite RNNs have been shown to be capable of simulating a universal Turing machine (Neto et al., 1997). Funahashi and Nakamura (1993) showed that the hidden state of a RNN can approximately represent dynamical systems of the same or less dimensional complexity. In particularly similar work, Rabusseau et al. (2018) showed that second order RNNs with linear activation functions are expressively equivalent to weighted finite automata.

Recent work has also explored the relationship between RNN internals and DFAs through a variety of methods. Although there have been multiple attempts at having RNNs learn a DFA structure based on input languages generated from DFAs and push down automata (Firoiu et al., 1998; Gers and Schmidhuber, 2001; Giles et al., 1992; Miclet and de la Higuera, 1996; Omlin and Giles, 1996a), most work has focused on *extracting* a DFA from the hidden states of a learned RNN. Early work in this field (Giles et al., 1991) demonstrated that grammar rules for regular grammars could indeed be extracted from a learned RNN. Other studies (Omlin and Giles, 1996b) tried to directly extract a DFA structure from the internal space of the RNN, often by clustering the hidden state activations from input stimuli, noting the transitions from one state to another given a particular new input stimuli. Clustering was done by a series of methods, such as K-Nearest Neighbor (Das and Mozer, 1993), K-means (Krakovna and Doshi-Velez, 2016), and Density Based Spatial Clustering of Applications with Noise (DBSCAN) (Das and Mozer, 1993; Lawrence et al., 2000). Another extraction effort (Ayache et al., 2018) uses spectral algorithm techniques to extract weighted automata from RNNs. Most recently, Weiss et al. (2018) have achieved state-of-the-art accuracy in DFA extraction by utilizing the L* query learning algorithm. Our work is different from these efforts in that we directly relate the RNN to a ground-truth minimal DFA, rather than extracting a machine from the RNN's state space.

The closest piece of related work is by Tiño et al. (1998). Like our work, this seeks to relate a RNN state with the state of a DFA. However, the RNN in Tiño et al. (1998) exactly mimics the DFA; also, the study is carried out in the context of a few specific regular languages that are recognized by automata with 2-3 states. In contrast, our work does not require exact behavioral correspondence between RNNs and DFAs: DFA states are allowed to be *abstracted*, leading to loss of information. Also, in our approach the mapping from RNN states to FA states can be approximate, and the accuracy of the mapping is evaluated quantitatively. We show that this allows us to establish connections

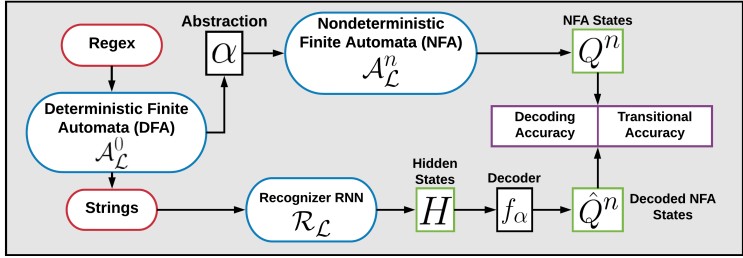

Figure 2: An overview of the state comparison experimental setup.

between RNNs and DFAs in the setting of a broad class of regular languages that often demand significantly larger automata (with up to 14 states) than those studied by Tiño et al. (1998).

## 3 DEFINITIONS

We start by introducing some definitions and notation. A *formal language* is a set of strings over a finite alphabet $\Sigma$ of input symbols. A *Deterministic Finite Automaton (DFA)* is a tuple $\mathcal{A} = (Q, \Sigma, \delta, q_0, F)$ where $Q$ is a finite set of states, $\Sigma$ is a finite alphabet, $\delta : Q \times \Sigma \to Q$ is a deterministic transition function, $q_0 \in Q$ is a starting state and $F \subset Q$ is a set of accepting states. $\mathcal{A}$ reads strings over $\Sigma$ symbol by symbol, starting from the state $q_0$ and making state transitions, defined by $\delta$, at each step. It accepts the string if it reaches a final accepting state in $F$ after reading it to the end. The set of strings accepted by a DFA is a special kind of formal language, known as a *regular language*. A regular language $\mathcal{L}$ can be accepted by multiple DFAs; such a DFA $\mathcal{A}_{\mathcal{L}}$ is *minimal* if there exists no other DFA $\mathcal{A}' \neq \mathcal{A}_{\mathcal{L}}$ such that $\mathcal{A}'$ exactly recognizes $\mathcal{L}$ and has fewer states than $\mathcal{A}_{\mathcal{L}}$. It can be shown that this minimal DFA (MDFA), which we denote by $\mathcal{A}_{\mathcal{L}}^0$, is *unique* (Hopcroft and Ullman, 1979).

**Abstractions.** A *Nondeterministic Finite Automaton (NFA)* is similar to a DFA, except that the deterministic transition function $\delta$ is now a non-deterministic transition relation $\delta^{NFA}$. This means that for a state $q$ in the NFA and $a \in \Sigma$, we have that $\delta^{NFA}(q, a)$ is now a subset of NFA states.

For a given regular language $\mathcal{L}$ we denote by $\mathcal{A}_{\mathcal{L}}^n$ a *Nondeterministic Finite Automaton (NFA)* with $n$ states that recognizes a superset of the language $\mathcal{L}$. An *abstraction map* is a map $\alpha : Q \to 2^Q$ that combines two states in NFA $\mathcal{A}_{\mathcal{L}}^n$, resulting in an NFA $\mathcal{A}_{\mathcal{L}}^{n+1}$; that is, $\mathcal{A}_{\mathcal{L}}^n \xmapsto{\alpha} \mathcal{A}_{\mathcal{L}}^{n+1}$. Since every DFA is also an NFA, we can apply $\alpha$ to the MDFA $\mathcal{A}_{\mathcal{L}}^0$ to obtain a NFA $\mathcal{A}_{\mathcal{L}}^1$. Intuitively, $\alpha$ creates a new NFA by combining two states of an existing NFA into a new 'superstate'. An NFA $\mathcal{A}_{\mathcal{L}}^n$ is an *abstraction* of $\mathcal{A}_{\mathcal{L}}^0$, if $\mathcal{A}_{\mathcal{L}}^n$ can be obtained from $\mathcal{A}_{\mathcal{L}}^0$ by repeated application of an abstraction map $\alpha$. Every state of $\mathcal{A}_{\mathcal{L}}^n$ can be viewed as a set of the states of the MDFA $\mathcal{A}_{\mathcal{L}}^0$, i.e $q^n \in Q^n \implies q^n = \{q_i^0\}_{i \in \mathcal{I}}$ with $q_i^0 \in Q^0$ for all $i$.

We define the *coarseness* of an abstraction $\mathcal{A}_{\mathcal{L}}^n$, as the number of applications of $\alpha$ on the MDFA required to arrive at $\mathcal{A}_{\mathcal{L}}^n$. Intuitively, repeated applications of $\alpha$ create NFAs that accept *supersets* of the language $\mathcal{L}$ recognized by the MDFA, and can hence be seen as coarse-grained versions of the original MDFA. The coarsest NFA, given by $\mathcal{A}_{\mathcal{L}}^{(|Q^0|-1)}$, is a NFA with only one accepting node and it accepts all strings on the alphabet $\Sigma$.

Given a regular language $\mathcal{L}$, we define $\mathcal{R}_{\mathcal{L}}$ to be a RNN that is trained to recognize the language $\mathcal{L}$, with a certain threshold accuracy. Each RNN $\mathcal{R}_{\mathcal{L}}$ will have a corresponding set of hidden states denoted by $H$. More details about the RNN are provided in §4.1. Note that a RNN can also be viewed as a transition system with 5-tuple $\mathcal{R} = (H, \Sigma, \delta^{\mathcal{R}}, h_0, F^{\mathcal{R}})$, where $H$ is a set of possible 'hidden' states (typically $H \subset \mathbb{R}^K$), $\delta^{\mathcal{R}}$ is the transition function of the trained RNN, $h_0$ is the initial state of the RNN, and $F^{\mathcal{R}}$ is the set of accepting RNN states. The key distinction between a DFA and a RNN is that the latter has a continuous state space, endowed with a topology and a metric.

**Decoding DFA States from RNNs.** Inspired by methods in computational neuroscience (Astrand. et al., 2014), we can define a *decoding function* or *decoder* $f : H \to Q^0$ as a function from the hidden states of a RNN $\mathcal{R}_{\mathcal{L}}$ to the states of the corresponding (for $\mathcal{L}$) MDFA $\mathcal{A}_{\mathcal{L}}^0 = (Q^0, \Sigma^0, \delta^0, q_0^0, F^0)$. We are interested in finding decoding functions that provide an insight into the internal knowledge representation of the RNN, which we quantify via the decoding and transitional accuracy metrics defined below.

**Decoding Abstraction States.** Let $\mathcal{A}_{\mathcal{L}}^n$ be an abstraction of $\mathcal{A}_{\mathcal{L}}^0$, obtained by applying $\alpha$ to $\mathcal{A}_{\mathcal{L}}^0$ repeatedly $n$ times, and let $Q^n$ be the set of states of $\mathcal{A}_{\mathcal{L}}^n$. We can define an *abstraction decoding function* $\hat{f} : H \to Q^n$, by $\hat{f}(h) := (\alpha^{|n|} \circ f)(h)$, that is the composition of $f$ with $\alpha^{|n|}$. The function $\alpha^{|n|}$ is the function obtained by taking $n$ compositions of $\alpha$ with itself. Given a dataset of input strings $\mathcal{D} \subset \Sigma^*$, we can define the *decoding accuracy* of a map $\hat{f}$ for an abstraction $\mathcal{A}_{\mathcal{L}}^n$ from RNN $\mathcal{R}_{\mathcal{L}}$ by:

$$\rho_{\hat{f}}(\mathcal{R}_{\mathcal{L}}, \mathcal{A}_{\mathcal{L}}^n) = \frac{1}{|\mathcal{D}|} \sum_{w \in \mathcal{D}} \sum_{t=0}^{|w|-1} \frac{\mathbb{1}(\hat{f}(h_{t+1}) = \alpha^{|n|}(q_{t+1}))}{|w|}$$

where $\mathbb{1}(C)$ is the boolean indicator function that evaluates to 1 if condition $C$ is true and to 0 otherwise, $h_{t+1} = \delta^{\mathcal{R}}(h_t, a_t)$ and $q_{t+1} = \delta^0(q_t, a_t)$. Note in particular, that for decoding abstraction states the condition is only checking if the $(t+1)$ RNN state is mapped to the $(t+1)$ NFA state by $\hat{f}$, which may be true even if the $(t+1)$ RNN state is not mapped to the $(t+1)$ MDFA state by the decoding function $f$. Therefore a function $\hat{f}$ can have a high decoding accuracy even if the underlying $f$ does not preserve transitions.

**Decoding Abstract State Transitions.** We now define an accuracy measure that takes into account how well transitions are preserved by the underlying function $f$.

Intuitively, for a given decoding function $\hat{f}$ and NFA $\mathcal{A}_{\mathcal{L}}^n$, we want to check whether the RNN transition on $a$ is mapped to the abstraction of the MDFA transition on $a$. We note that in the definition of the decoding function, we take into account only the states at $(t+1)$ and not the underlying transitions in the original MDFA $\mathcal{A}_{\mathcal{L}}^0$, unlike we do here. More precisely, the *transitional accuracy* of a map $\hat{f}$ for a given RNN and abstraction, with respect to a data-set $\mathcal{D}$, is defined as:

$$\phi_{\hat{f}}(\mathcal{R}_{\mathcal{L}}, \mathcal{A}_{\mathcal{L}}^n) = \frac{1}{|\mathcal{D}|} \sum_{w \in \mathcal{D}} \sum_{t=0}^{|w|-1} \frac{\mathbb{1}(\hat{f}(\delta^{\mathcal{R}}(h_t, a_t)) = \alpha^{|n|}(\delta^0(f(h_t), a_t)))}{|w|}$$

Our experiments in the next section demonstrate that decoding functions with high decoding and transitional accuracies exist for abstractions with relatively low coarseness.

## 4 EXPERIMENTAL RESULTS

Our goal is to experimentally test the hypothesis that a high accuracy, low coarseness decoder exists from $\mathcal{R}_{\mathcal{L}}$ to $\mathcal{A}_{\mathcal{L}}^0$. We aim to answer 4 fundamental questions related to the transitional accuracy of $\mathcal{A}_{\mathcal{L}}^0$ and $\mathcal{R}_{\mathcal{L}}$: (1) *How do we choose an appropriate abstraction decoding function $\hat{f}$?* (2) *What necessitates the abstraction function $\alpha$?* (3) *Can we verify that a low coarseness $\alpha$ and high accuracy $\hat{f}$ exists?* and lastly, (4) *How can we better understand $\mathcal{R}_{\mathcal{L}}$ in the context of $\alpha$ and $\hat{f}$?*

### 4.1 EXPERIMENTAL DESIGN

To answer the above questions and evaluate our claims with statistical significance, we have designed a flexible framework that facilitates comparisons between states of $\mathcal{A}_{\mathcal{L}}^0$ and $\mathcal{R}_{\mathcal{L}}$, as summarized in Figure 2.

We first randomly generate a regular expression specifying language $\mathcal{L}$ with MDFA $\mathcal{A}_{\mathcal{L}}^0$. Using $\mathcal{A}_{\mathcal{L}}^0$, we randomly generate a training dataset $\mathcal{D} \equiv \mathcal{D}_+ \bigcup \mathcal{D}_-$ of positive ($x_+ \in \mathcal{L}$) and negative ($x_- \notin \mathcal{L}$) example strings (see Appendix for details). We then train $\mathcal{R}_{\mathcal{L}}$ with $\mathcal{D}$ on the language recognition task: given an input string $x \in \Sigma^*$, is $x \in \mathcal{L}$? Thus, we have two language recognition models corresponding to state transition systems from which state sequences are extracted. Given a length $T$ input string $\mathbf{x} = (x_1, x_2, x_t, ..., x_T) \in \mathcal{D}$, let the categorical states generated by $\mathcal{A}_{\mathcal{L}}^0$ be denoted by $\mathbf{q} = (q_0, q_1, q_t, ..., q_T)$ and continuous states generated by the $\mathcal{R}_{\mathcal{L}}$ be $\mathbf{h} = (h_0, h_1, h_t, ..., h_T)$. The recorded state trajectories ($\mathbf{q}$ and $\mathbf{h}$) for all input strings $\mathbf{x} \in \mathcal{D}$ are used as inputs into our analysis.. For our experiments, we sample a total of $\sim 500$ unique $\mathcal{L}$, and thus perform an analysis of $\sim 500$ recognizer MDFAs and $\sim 500$ trained recognizer RNNs.

### 4.2 LEARNING AN ACCURATE DECODER

As mentioned in the beginning of §4, we must first determine what is a reasonable form for the decoders $f$ and $\hat{f}$ to ensure high accuracy on the decoding task. Figure 3b shows decoding accuracy

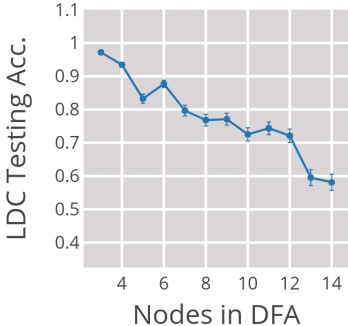 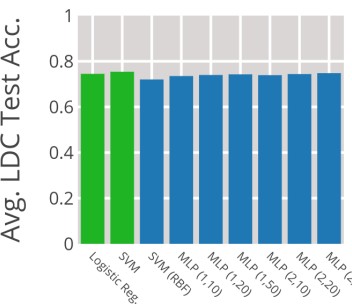

Figure 3: a (Left): Average linear decoding accuracy as a function of $M$. 3b (Right): Average decoding accuracy showing no statistically significant difference between linear (green) and nonlinear (blue) decoders for all MDFAs tested.

$\mathbb{E}_{\mathcal{D}}[\rho_f(\mathcal{R}, \mathcal{A}_{\mathcal{L}}^0)|f]$ for several different decoding functions $f$. We test two linear classifiers (Multinomial Logistic Regression and Linear Support Vector Machines (SVM)) and two non-linear classifiers (SVM with a RBF kernel, Multi-Layer Perceptrons with varying layers and hidden unit sizes). In order to evaluate whether accuracy varies significantly amongst all decoders, we use a statistically appropriate F-test. Surprisingly, we find there to be no statistical difference umong our sampled languages: the nonlinear decoders achieve no greater accuracy than the simpler linear decoders. We also observe in our experiments that as the size of the MDFA $M$ increases, the decoding accuracy decreases for all decoders in a similar manner. Figure 3a shows this relationship for the multinomial logistic regression classifier.

Taken together, these results have several implications. First, we find that a highly expressive nonlinear decoder does not yield any increase in decoding accuracy, even as we scale up in MDFA complexity. We can conclude from this finding and our extensive hyperparameter search for each decoder model that the decoder models we chose are expressive enough for the decoding task. Second, we find that decoding accuracy for MDFA states is in general not very high. These two observations suggest linear decoders are sufficient for the decoding task, but also suggests the need for a different interpretation of the internal representation of the trained RNN.

### 4.3 WHY ABSTRACTIONS ARE NECESSARY

Given the information above, how is the hidden state space of the $\mathcal{R}_{\mathcal{L}}$ organized? One hypothesis that is consistent with the observations above is that the trained RNN reflects a coarse-grained abstraction of the state space $\mathcal{Q}^0$ (Figure 1), rather than the MDFA states themselves.[1]

To test this hypothesis, we propose a simple greedy algorithm to find an abstraction mapping $\alpha$: (a) given an NFA $A_{\mathcal{L}}^n$ with $n$ unique states in $\mathcal{Q}^n$, consider all $(n-1)$-partitions of $\mathcal{Q}^{n-1}$ (i.e. two NFA states $s, s'$ have merged into a single superstate $\{s, s'\}$); (b) select the partition with the highest decoding accuracy; (c) Repeat this iterative merging process until only a 2-partition remains. We note that this algorithm does not *explicitly* take into consideration the transitions between states which are essential to evaluating $\phi_{\hat{f}}(\mathcal{R}_{\mathcal{L}}, \mathcal{A}_{\mathcal{L}}^n)$. Instead, the transitions are taken into account *implicitly* while learning the decoder $f$ at each iteration of the abstraction algorithm. Decreasing the number of states in a classification trivially increases $\mathbb{E}_{\mathcal{D}}[\rho_f(\mathcal{R}, \mathcal{A}_{\mathcal{L}}^n)|f]$. We compare to a baseline where the states abstracted are random to validate our method. We compute the normalized Area Under the Curve (AUC) of a decoder accuracy vs coarseness plot. Higher normalized AUC indicates a more accurate abstraction process. We argue through Figure 4a that our method gives a non-trivial increase over the abstraction performance of a random baseline.

The abstraction algorithm is greedy in the sense that we may not find the globally optimal partition (i.e. with the highest decoding accuracy and lowest coarseness), but an exhaustive search over all partitions is computationally intractable. The greedy method we have proposed has $\mathcal{O}(M^2)$ complexity instead, and in practice gives satisfactory results. Despite it being greedy, we note that the resulting sequence of clusterings are stable with respect to randomly chosen initial conditions and model parameters. Recognizer RNNs with a different number of hidden units result in clustering sequences that are consistent with each other in the critical first few abstractions.

---

[1]This idea can be motivated by recasting the regular expression for (say) E-Mails into a hierarchical grammar with production rules.

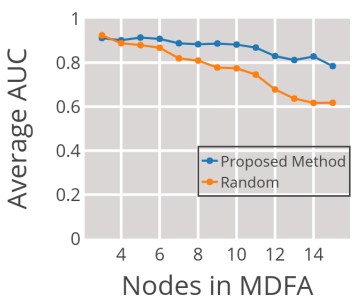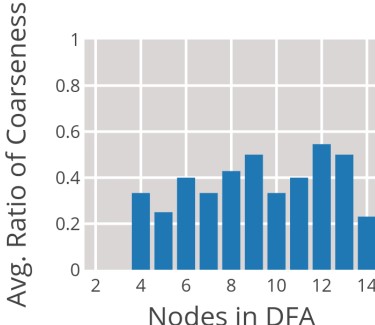

Figure 4: 4a (Left) Average normalized Area under the curve (AUC) for all decoding accuracy vs coarseness plots (similar to Figure 8). 4b (Right): Average ratio of coarseness that must be created relative to $M$ in the MDFA to achieve $90\%$ testing accuracy.

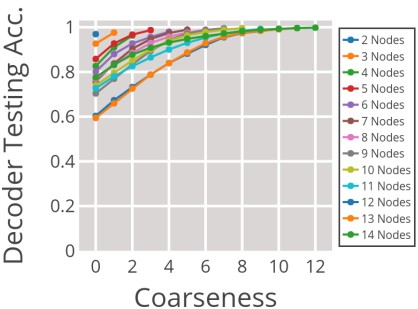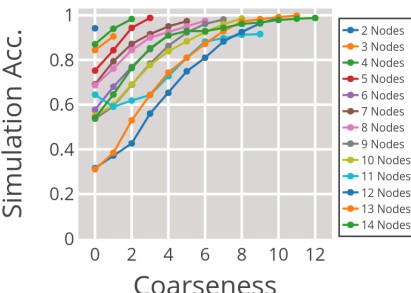

Figure 5: 5a (Left): Average linear decoder testing accuracy as a function of coarseness (The number of times $\alpha$ is applied), sorted by the number of nodes in the MDFA. 5b (Right): Average transitional accuracy vs. coarseness, sorted by the number of nodes in the MDFA while using a linear decoder.

## 4.4 DECODING ABSTRACTIONS AND TRANSITIONS

Once an abstraction $\alpha$ has been found, we can evaluate whether the learned abstraction decoder $\hat{f}$ is of high accuracy, and whether the $\alpha$ found is of low coarseness. Results showing the relationship between high decoding accuracy $\rho_{\hat{f}}(\mathcal{R}_\mathcal{L}, \mathcal{A}_\mathcal{L}^n)$ as a function of coarseness is presented in Figure 5a conditioned on the number of nodes in the original MDFA. As stated in §4.2, as $M$ increases, $\rho_{\hat{f}}(\mathcal{R}_\mathcal{L}, \mathcal{A}_\mathcal{L}^n)$ decreases on the MDFA (i.e. $n = 0$). We attribute this to two factors, (1) as $M$ increases, the decoding problem naturally increases in difficulty, and (2) $\mathcal{R}_\mathcal{L}$ abstracts multiple states of $\mathcal{A}_\mathcal{L}$ into a single state in $H$ as can be seen empirically from Figure 1. We validate the second factor by training a overparameterized non-linear decoder on the decoding task and find no instances where the decoder obtains $0\%$ training error. Alongside the decoding accuracy, we also present transitional accuracy $\phi_{\hat{f}}(\mathcal{R}_\mathcal{L}, \mathcal{A}_\mathcal{L}^n)$ as a function of coarseness Figure 5b. Both of these figures showcase that for a given DFA, in general we can find a low coarseness NFA that the hidden state space of $\mathcal{R}_\mathcal{L}$ can be decoded to with high accuracy. Figure 4b shows the average ratio of abstractions relative to $M$ needed to decode to 90% accuracy, indicating low coarseness relative to a random baseline. For completeness, we also present decoder and transition accuracy for a *nonlinear* decoder in Figures 6a and 6b showing similar results as the linear decoder.

Our fundamental work shows a large scale analysis of how RNNs $\mathcal{R}_\mathcal{L}$ relate to abstracted NFAs $\mathcal{A}_\mathcal{L}^n$ for hundreds of minimal DFAs, most of which are much larger and more complex than DFAs typically used in the literature. By evaluating the transition accuracy between $\mathcal{R}$ and $\mathcal{A}_\mathcal{L}^n$ we empirically validate our claim. We show that there does exist high accuracy decoders from $\mathcal{R}$ to an abstracted NFA $\mathcal{A}_\mathcal{L}^n$.

## 4.5 INTERPRETING THE RNN HIDDEN STATE SPACE WITH RESPECT TO THE MINIMAL DFA

With an established high accuracy $\hat{f}$ with low coarseness $\alpha$ reveals a unique interpretation of $H$ with respect to $\mathcal{A}_\mathcal{L}^0$. Using $\alpha$ and $f$ to relate the two, we uncover an interpretation of how $\mathcal{R}$ organizes $H$ with respect to $\mathcal{A}_\mathcal{L}^n \, \forall \, n \in [M]$. We can then determine the appropriate level of abstraction the network uses to accomplish the logical language recognition task in relation to the underlying MDFA. We provide two example 'real-world' DFAs to illustrate this interpretation and show several interesting patterns.

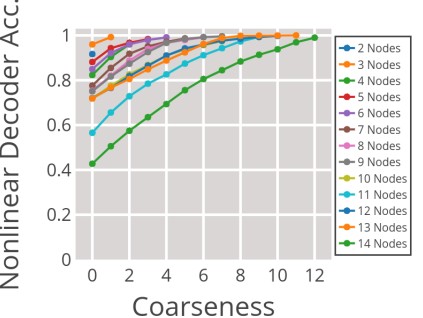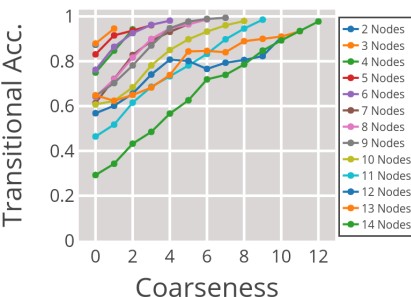

Figure 6: 6a (Left): Average nonlinear testing accuracy as a function of coarseness, sorted by the number of nodes in the MDFA. 6b (Right): Average transitional accuracy vs coarseness, sorted by the number of nodes in the MDFA while using a non-linear decoder.

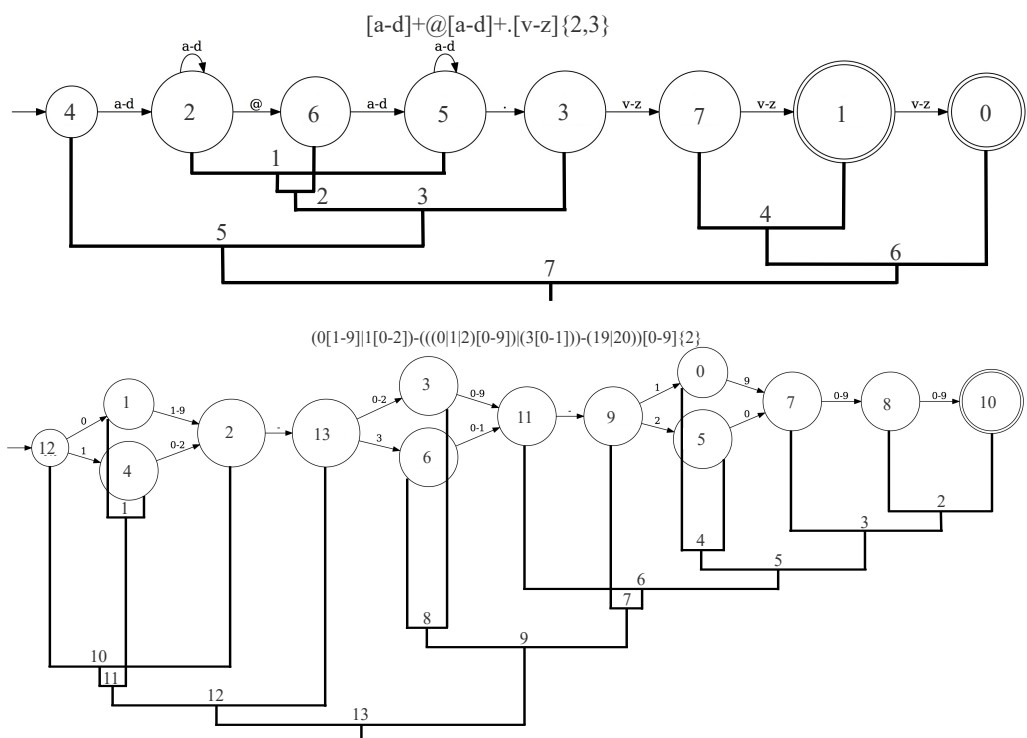

Figure 7: 7a (Top): The MDFA of the SIMPLE EMAILS language with a dendrogram representing the the sequence of abstractions created while using a linear decoder. Showing the initial abstractions are those of the same pattern [a-d]*. 7b (Bottom) The MDFA of the DATES language with a dendrogram representing the the sequence of abstractions created while using a linear decoder. Showing the initial abstractions are those representing states that represent the same moment in time.

We present in Figure 7 the clustering sequences of two regular expressions that have real-world interpretations, namely the SIMPLE EMAILS and DATES languages that recognize simple emails and simple dates respectively. To explain, Figure 7b shows the DATES language with its clustering sequence superimposed on the MDFA in the form of a dendrogram. The dendrogram can be read in a top-down fashion, which displays the membership of the MDFA states and the sequence of abstractions up to $n = M - 1$. A question then arises: *How should one pick a correct level of abstraction $n$?*. The answer can be seen in the corresponding accuracies $\rho_{\hat{f}}(\mathcal{R}_{\mathcal{L}}, \mathcal{A}_{\mathcal{L}}^n)$ in Figure 8.

As $n$ increases and the number of total NFA states decreases, the linear decoding (LDC) prediction task obviously gets easier (100% accuracy when the number of NFA states $Q^{|Q|-1}$ is 1), and hence it is important to consider how to choose the number of abstractions in the final partition. We typically set a threshold for $\rho_{\hat{f}}(\mathcal{R}_{\mathcal{L}}, \mathcal{A}_{\mathcal{L}}^n)$ and select the minimum $n$ required to achieve the threshold accuracy.

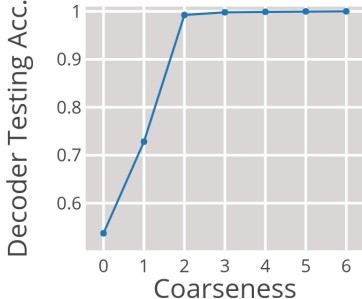 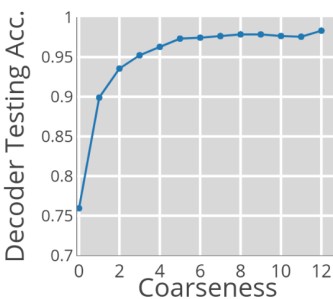

Figure 8: 8a (Left): Linear decoder accuracies as a function of coarseness for the SIMPLE EMAILS language in Figure 7a. 8b (Right): Linear decoder accuracies as a function of coarseness for the DATES language corresponding to Figure 7b.

Consider the first two abstractions of the SIMPLE EMAILS DFA. We notice that both states 2 and 5 represent the pattern matching task `[a-d]*`, because they are agglomerated by the algorithm. Once two abstractions have been made, the decoder accuracy is at a sufficient point, as seen in Figure 8. This suggests that the collection of hidden states for the two states are not linearly separable. One possible and very likely reason for this is the network has learned an abstraction of the pattern `[a-d]*` and uses the same hidden state space regardless of location in string to recognize this pattern, which has been indicated in past work (Karpathy et al., 2015). This intuitive example demonstrates the RNN's capability to learn and abstract patterns from the DFA. This makes intuitive sense because $\mathcal{R}_\mathcal{L}$ does not have any direct access to $\mathcal{A}_\mathcal{L}^0$, only to samples generated from $\mathcal{A}_\mathcal{L}^0$. The flexibility of RNNs allows such abstractions to be created easily.

The second major pattern that arises can be seen in the dendrogram in the bottom row of Figure 7. We notice that, generally, multiple states that represent the same location in the input string get merged (1 and 4, 3 and 6, 0 and 5). The SIMPLE EMAILS dendrogram shows patterns that are location-independent, while the fixed length pattern in the DATES regex shows location-dependent patterns. We also notice that the algorithm tends to agglomerate states that are within close sequential proximity to each other in the DFA, again indicating location-dependent hierarchical priors. Overall, our new interpretation of $H$ reveals some new intuitions, empirically backed by our decoding and transitional accuracy scores, regarding how the RNN $\mathcal{R}_\mathcal{L}$ structures the hidden state space $H$ in the task of language recognition. We find patterns such as these in almost all of the DFA's tested. We provide five additional random DFA's in the appendix (Figures 9–13) to show the wide variability of the regular expressions we generate/evaluate on.

## 5 CONCLUSIONS

We have studied how RNNs trained to recognize regular formal languages represent knowledge in their hidden state. Specifically, we have asked if this internal representation can be decoded into canonical, minimal DFA that exactly recognizes the language, and can therefore be seen to be the "ground truth". We have shown that a linear function does a remarkably good job at performing such a decoding. Critically, however, this decoder maps states of the RNN not to MDFA states, but to states of an *abstraction* obtained by clustering small sets of MDFA states into "abstractions". Overall, the results suggest a strong structural relationship between internal representations used by RNNs and finite automata, and explain the well-known ability of RNNs to recognize formal grammatical structure.

We see our work as a fundamental step in the larger effort to study how neural networks learn formal logical concepts. We intend to explore more complex and richer classes of formal languages, such as context-free languages and recursively enumerable languages, and their neural analogs.

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

## A    DATASET GENERATION

In order to generate a wide variety of strings that are both accepted and rejected by the DFA corresponding to a given regex $R$, we use the Xeger Java library, built atop the dk.brics.automaton library Møller (2017). The Xeger library, given a regular expression, generates strings that are accepted by the regular expression's corresponding DFA. However, there is no standard method to generate examples that would be rejected by the DFA. These rejected examples need to be diverse to properly train an acceptor/rejector model: if the rejected examples are completely different from the accepted examples, the model will not be able to discern between similar input strings, even if one is an accepted string and the other is a rejected string. However, if the rejected examples were too similar to the accepted examples, the model would not be able to make a judgment on a completely new string that does not resemble any input string seen during training. In other words we want the rejected strings to be drawn from two distinct distributions, one similar and one independent compared to the distribution of the accepted strings. In order to achieve this, we generate negative examples in two ways: First, we randomly swap two characters in an accepted example enough times until we no longer have an accepted string. And secondly, we take an accepted string and randomly shuffle the characters, adding it to our dataset if the resulting string is indeed rejected.

In our experiments we generate 1000 training examples with a 50:50 accept/reject ratio. When applicable we generate strings of varying length capped at some constant, for example with the SIMPLE EMAILS language we generate strings of at most 20 characters.

### A.1    EXAMPLE REGULAR EXPRESSIONS, CORRESPONDING DFAS, AND HIERARCHIES DECODED FROM OUR FRAMEWORK

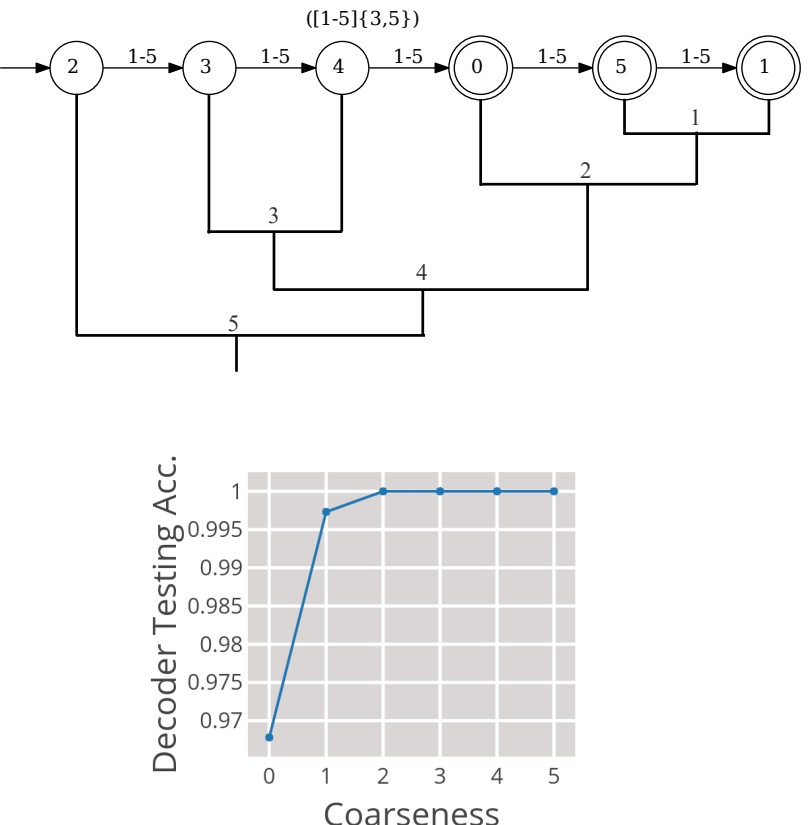

Figure 9: (Top): Typical regular expression and corresponding DFA generated by our framework. A dendrogram superimposed on the DFA shows the hierarchy of the RNN's hidden state space. (Bottom): Linear decoding accuracy as a function of coarseness corresponding to DFA above.

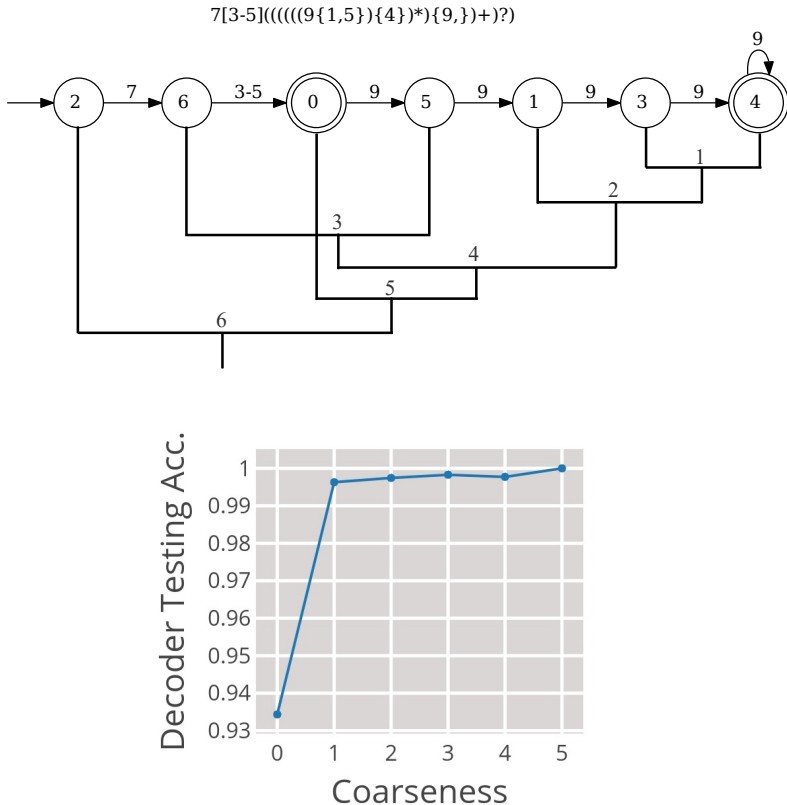

Figure 10: (Top): Typical regular expression and corresponding DFA generated by our framework. A dendrogram superimposed on the DFA shows the hierarchy of the RNN's hidden state space. (Bottom): Linear decoding accuracy as a function of coarseness corresponding to the DFA above.

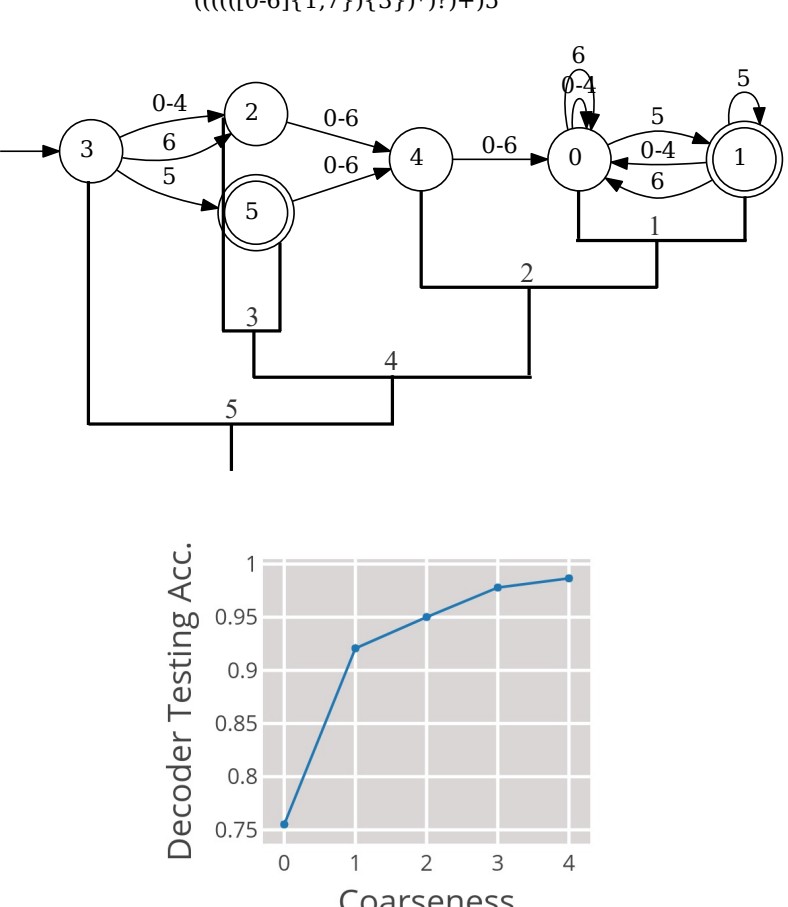

Figure 11: (Top): Typical regular expression and corresponding DFA generated by our framework. A dendrogram superimposed on the DFA shows the hierarchy of the RNN's hidden state space. (Bottom): Linear decoding accuracy as a function of coarseness corresponding to the DFA above.

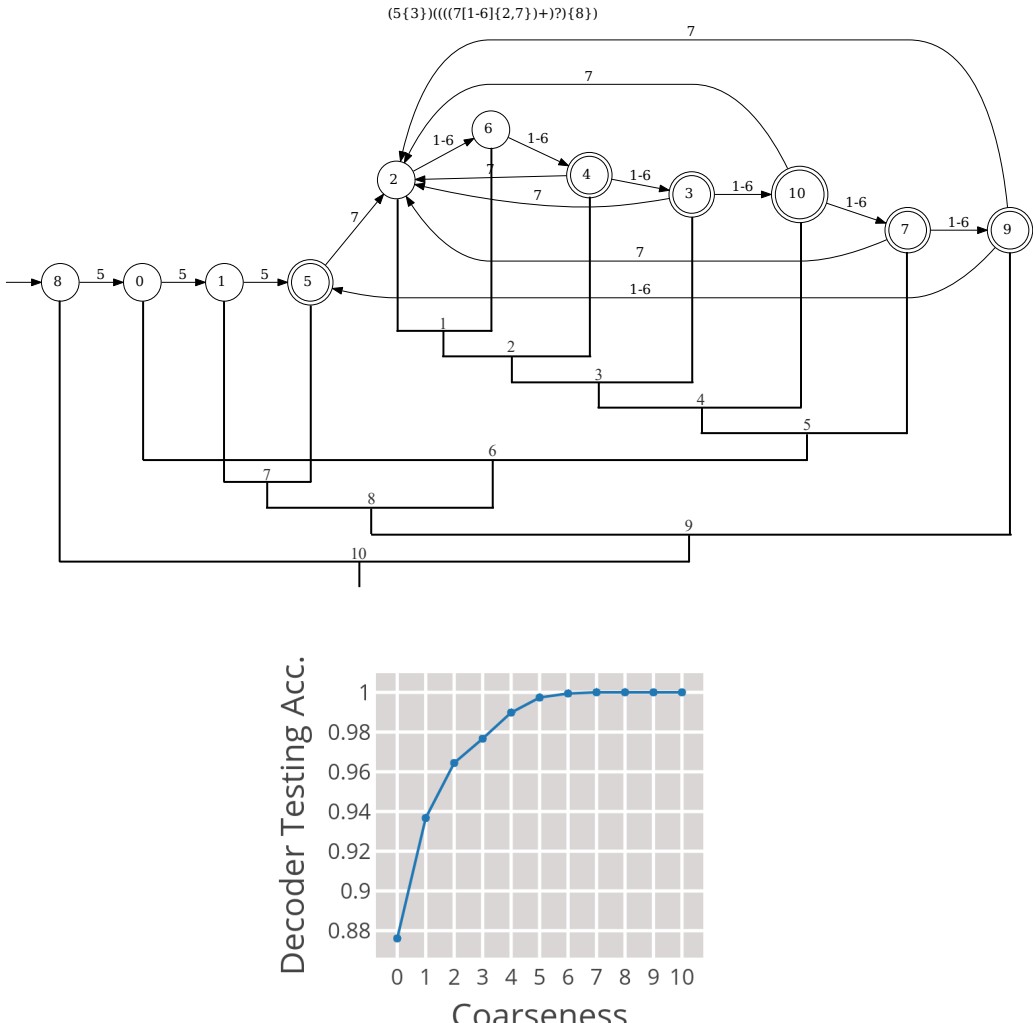

Figure 12: (Top): Typical regular expression and corresponding DFA generated by our framework. A dendrogram superimposed on the DFA shows the hierarchy of the RNN's hidden state space. (Bottom): Linear decoding accuracy as a function of coarseness corresponding to the DFA above.

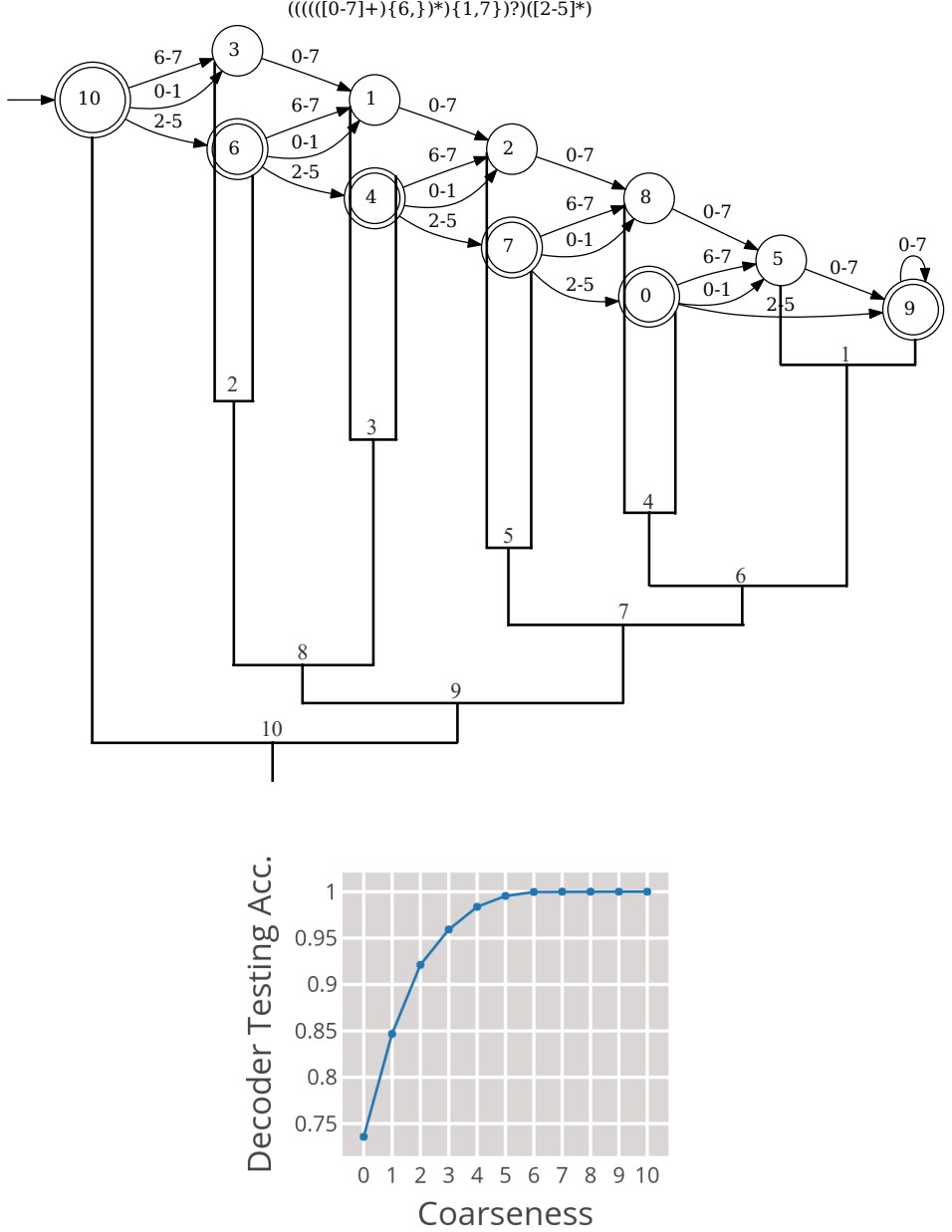

Figure 13: (Top): Typical regular expression and corresponding DFA generated by our framework. A dendrogram superimposed on the DFA shows the hierarchy of the RNN's hidden state space. (Bottom): Linear decoding accuracy as a function of coarseness corresponding to the DFA above.

