# OpenReview forum: "Representing Formal Languages: A Comparison Between Finite Automata and Recurrent Neural Networks "
_ICLR.cc/2019/Conference_

### Official Review · AnonReviewer3 · 2018-11-01
**Well written paper -- One major concern**

**Rating:** 5
**Confidence:** 3

**Review:**

Paper Summary -
The authors trained RNNs to recognize formal languages defined by random regular expressions, then measured the accuracy of decoders that predict states of the minimal deterministic finite automata (MDFA) from the RNN hidden states. They then perform a greedy search over partitions of the set of MDFA states to find the groups of states which, when merged into a single decoder target, maximize prediction accuracy. For both the MDFA and the merged classes prediction problems, linear decoders perform as well as non-linear decoders.
Clarity - The paper is very clear, both in its prose and maths.
Originality - I don't know of any prior work that approaches the relationship between RNNs and automata in quite this way.
Quality/Significance - I have one major concern about the interpretation of the experiments in this paper.

The paper seems to express the following logic:
1 - linear (and non-linear) decoders aren't so good at predicting MDFA states from RNN hidden states
2 - if we make an "abstract" finite automata (FA) by merging states of the MDFA to optimize decoder performance, the linear (and non-linear) decoders are much better at predicting this new, smaller FA's states.
3 - thus, trained RNNs implement something like an abstract FA to recognize formal languages.

However, a more appropriate interpretation of these experiments seems to be:
1 - (same)
2 - if we find the output classes the decoder is most often confused between, then merge them into one class, the decoder's performance increases -- trivially. in other words, you just removed the hardest parts of the classification problem, so performance increased. note: performance also increases because there are fewer classes in the merged-state FA prediction problem (e.g., chance accuracy is higher).
3 - thus, from these experiments it's hard to say much about the relationship between trained RNNs and finite automata.

I see that the "accuracy" measurement for the merged-state FA prediction problem, \rho, is somewhat more complicated than I would have expected; e.g., it takes into account \delta and f(h_t) as well as f(h_{t+1}). Ultimately, this formulation still asks whether any state in the merged state-set that contains f(h) transitions under the MDFA to the any state in the merged state-set that contains f(h_{t+1}). As a result, as far as I can tell the basic logic of the interpretation I laid out still applies.

Perhaps I've missed something -- I'll look forward to the author response which may alleviate my concern.

Pros - very clearly written, understanding trained RNNs is an important topic
Cons - the basic logic of the conclusion may be flawed (will await author response)

Minor -
The regular expression in Figure 6 (Top) is for phone numbers instead of emails.
"Average linear decoding accuracy as a function of M in the MDFA" -- I don't think "M" was ever defined. From contexts it looks like it's the number of nodes in the MDFA.
"Average ratio of coarseness" -- It would be nice to be explicit about what the "ratio of coarseness" is. I'm guessing it's (number of nodes in MDFA)/(number of nodes in abstracted DFA).
What are the integers and percentages inside the circles in Figure 6?
Figures 4 and 5 are difficult to interpret because the same (or at least very similar) colors are used multiple times.
I don't see "a" (as in a_t in the equations on page 3) defined anywhere. I think it's meant to indicate a symbol in the alphabet \Sigma. Maybe I missed it.

---

> ### Author Response · Authors · 2018-11-26
> **Thank you for the in-depth review**
>
> We thank the reviewer for a careful and thorough review of our paper.
>
> It is true that in a classification problem, if you merge the most confused classes together, classification accuracy increases trivially. However, our paper’s intention was to not make a logical connection between RNNs and automata based on this observation, but rather to show that on a per-example basis, the most confused states that are merged reveal geometric interpretations behind how the RNN encodes the MDFA. By analyzing the accuracy vs coarseness curves (Figure 7) alongside the dendrograms (Figure 6) for two regular expressions that have a real-world interpretation, we gain a novel interpretation of the similarity between the internal representation of the RNN and the MDFA. We consistently find that MDFA states that are linearly inseparable by the decoder often refer to the same pattern in the original regular expression. Our abstraction method provides an interpretable relationship between these two states as evidenced by our dendrograms. We provide two dendrogram results specifically for regular expressions with clear meaning to showcase these patterns. We will provide more dendrograms in the final version to show how consistent the patterns are.
>
> -Why is the definition of the “accuracy” measurement \rho more complicated than expected at first glance?
> The accuracy measure is a quantitative measure predicated on \delta, f(h_t) and f(h_{t+1}), because we need these mappings to capture the structural similarities between RNNs and the abstraction of the MDFA. The accuracy is an average of averages where we calculate the average over a dataset of strings D, with strings of varying lengths. For each individual string we are interested in the number of alphabets for which the decoding f(.) respects the transition \delta in the MDFA, when transitioning from h_t to h{t+1}.
>
> We agree with all of the minor comments and clarity concerns that the reviewer has and will address them in the final version of our paper.

---

> > ### Comment · Area_Chair1 · 2018-11-30
> > **Reviewer 3 please consider this response**
> >
> > Reviewer 3, thank you for your review. Does the author response (above) address your major concern? If not, please take the remaining few days to follow on in this discussion. If you are in a position to reconsider your assessment, please do so, and if you stand by your score, please provide a short explanation as to where the rebuttal falls short.

---

> > > ### Author Response · Authors · 2018-12-05
> > > **Please consider our latest response to reviewer 2**
> > >
> > > Reviewer 3, we have addressed many of your concerns in our response to reviewer 2 above. We would like to emphasize that there are 2 significant misunderstanding about the core of our logical conclusions of our paper. We have clarified them in our response to reviewer 2 above. We ask that reviewer 2, reviewer 3, and the area chair please consider these clarifications as we believe that they will significantly affect the evaluation of our work.

---

### Official Review · AnonReviewer2 · 2018-11-02
**Interesting idea, serious clarity problems**

**Rating:** 5
**Confidence:** 3

**Review:**

This paper aims to show that an RNN trained to recognize regular languages effectively focuses on a more abstract representation of the FSA of the corresponding language.

Understanding the type of information encoded in the hidden states of RNNs is an important research question. Recent results have shown connections between existing RNN architectures and both weighted (e.g., Chen et al., NAACL 2018, Peng et al., EMNLP 2018) and unweighted (Weiss et al., ACL 2018) FSAs. This paper asks a simple question: when trained to recognize regular languages, do RNNs converge on the same states as the corresponding FSA? While exploring solutions to this question is potentially interesting, there are significant clarity issues in this paper which make it hard to understand it. Also, the main claim of the paper — that the RNN is focusing on a low level abstraction of thew FSA — is not backed-up by the results.

Comments:

— The authors claim that the RNN states map to FSA states with *low* coarseness, but Figure 3b (which is never referred to in text…) shows that in most cases the ratio of coarseness is at least 1/3, and in some cases > 1/2.

— Clarity:
While the introduction is relatively clear starting from the middle of section 3 there are multiple clarity issues in this paper. In the current state of affairs it is hard for me to evaluate the full contribution of the paper.

- The definitions in section 3 were somewhat confusing. What is the conceptual difference between the two accuracy definitions?

- When combining two states, does the new FSA accept most of the strings in the original FSAs? some of them? can you quantify that? Also, figure 6 (which kind of addresses this question) would be much more helpful if it used simple expressions, and demonstrated how the new FSA looks like after the merge.

- section 4 leaves many important questions unanswered:
1. Which RNN was used? which model? which parameters? which training regime? etc.
2. How were the expressions sampled? the authors mention that they were randomly sampled, so how come they talk about DATE and EMAIL expressions?
3. What is the basic accuracy of the RNN classifier (before decoding)? is it able to learn to recognize the language? to what accuracy?

- Many of the tables and figures are never referred to in text (Figure 3b, Figure 5)

- In Figure 6, there is a mismatch between the regular expression (e.g., [0-9]{3}….) and the transitions on the FSA (a-d, @).

- How come Figure 3a goes up to 1.1? isn’t it bounded by 1? (100%?)

- The negative sampling procedure should be described in the main text, not the appendix. Also, it is not clear how come shuffling the characters is considered an independent distribution.

---

> ### Author Response · Authors · 2018-11-26
> **Thank you for the in-depth questions and comments 1/2**
>
> We thank the reviewer for the in-depth questions and comments, and look forward to any follow-up questions or concerns.
>
> -The authors claim that the RNN states map to FSA states with *low* coarseness, but Figure 3b (which is never referred to in text…) shows that in most cases the ratio of coarseness is at least 1/3, and in some cases > 1/2.
> We define coarseness to be “low” when the number of abstractions needed to reach 90% decoding accuracy, as in Figure 3b, is low relative to the number of abstractions needed to reach such a decoding accuracy when abstractions are formed randomly, as opposed to our greedy method of abstracting states. In figure 4a, the area under each plotted curve will be higher if the decoder is able to reach higher accuracies with a fewer number of abstractions (“lower coarseness”.) Following this logic, we have plotted the average area under the curve (AUC) for our strategy, along with the strategy of randomly abstracting states in the appendix of our paper. The added benefit of our method can be seen over random by the increase in average AUC for each collection of MDFAs with M states.  We show that the AUC is highest when employing our greedy strategy, indicating that the coarseness is indeed “low” with respect to other abstraction strategies.
>
> -What is the conceptual difference between the two accuracy definitions?
> The conceptual difference between decoding accuracy and transitional accuracy are two levels of abstraction to viewing the map \hat{f}. Decoding accuracy asks how well \hat{f} can map the RNN state to the abstracted NFA state, which is essentially asking a membership query, while preserving the MDFA transitions. Transitional accuracy asks if the mapping accurately preserves the transitions from state s_t to s_{t+1} on the given input a_t in the abstracted NFA. The decoding accuracy requires that the transitions of the MDFA are preserved by the mapping \hat{f}, while the transitional accuracy considers the transitions in the abstraction.
>
> -Which RNN was used? Which model? Which parameters? Which training regime?
> We performed an extensive hyperparameter search, varying number of hidden units and layers, mini-batch size, dropout rates, learning rates, and max number of training epochs. The best performing architecture -- one that is able to achieve high validation accuracies across the wide range of regular languages used in our framework -- is a 2 layer, 50 hidden unit vanilla RNN, trained via SGD for 100 epochs with a mini-batch size of 30, dropout probability of 0.4, and learning rate of 0.0003. The inputs to the model was optimized to predict a binary variable under a cross entropy loss. We will include these details in the final paper.
>
> -How were the regular expressions sampled?
> We randomly sample expressions using a probabilistic context free grammar based on the specification in the bk.brics.automata java documentation (http://www.brics.dk/automaton/doc/dk/brics/automaton/RegExp.html).Two examples are shown in the appendix of the expressions sampled by our framework. Our intention behind showing an EMAILS and DATES regular expressions that were formed outside of the aforementioned framework was to show how a typical, easily interpretable recognition algorithm is encoded by the RNN. We didn’t want the reader to be distracted by the regular expression itself but rather bring light to the interpretation of the dendrograms in section 4.5.
>
> For transparency and reproducibility, we will release the source code for our framework.
>
> -What is the basic accuracy of the RNN Recognizer?
> For a recognizer RNNs to be included in the decoding experiments, we required a minimum classification test accuracy of 95%. We will add this detail in the final version of the paper.

---

> > ### Comment · AnonReviewer2 · 2018-11-28
> > **Thank you for your response**
> >
> > As to the definition of “low”, the fact that it’s lower than a random baseline doesn’t mean that it’s absolutely *low* (this term is not even well-defined). I am not sure if this is that important, but it seems to be a major claim of this paper which is repeated several times and feels at the very least inaccurate. More importantly, this relates to AnonReviewer3’s concern: there is a simpler explanation to these results which the authors are not addressing. I have seen the authors’ response to this concern and was not convinced: if (to cite the authors’ response), “our paper’s intention was to not make a logical connection between RNNs and automata (…)”, then significant parts of the paper need to be re-written. Based on the response, the contribution of this paper largely relies on two cherry-picked examples.
> >
> > As to the response: “For a recognizer RNNs to be included in the decoding experiments, we required a minimum classification test accuracy of 95%.”:  which proportion of the cases meet this threshold?

---

> > > ### Author Response · Authors · 2018-12-04
> > > **Clarifying some significant misunderstandings**
> > >
> > > We believe there are two significant misunderstandings here. First, Reviewer 3 states “if we find the output classes the decoder is most often confused between, then merge them into one class, the decoder's performance increases -- trivially.” The word “trivially” is the problem here, as merging two classes that are easily confused by a highly trained classifier can actually be quite informative.  Consider the example of a trained face recognition classifier that easily confuses identical twins. If we merge Twin1 and Twin2 into a single new superclass Twins = {Twin1, Twin2} then the resulting classifier will certainly perform better and for good reason: the twins are highly related and thus have similar looks. Iterating this kind of confusion-based merging is a valid form of hierarchical clustering (e.g. merging plants together, and then animals together, etc. to learn a taxonomy).  In short, the increase in prediction accuracy after merging is “trivial”, but the interpretation for why an increase occurs is certainly not: finding classes that are easily confused is important information about the similarity metric learned by the classifier.
> > >
> > > The second misunderstanding involves our earlier response to Reviewer 3 where we state “our paper’s intention was to not make a logical connection between RNNs and automata (…).” This statement has been taken out of context. The critical part of that sentence is in the “(...)”, namely “based on this observation...”. Without that context, it seems like we are negating the core conclusion of our paper -- that there is indeed a connection between the hidden state space of the RNN and that of the MDFA. However, that was not our intent. We were just trying to convey that our conclusion is not based on that particular observation(“It is true that in a classification problem, if you merge the most confused classes together, classification accuracy increases...”); instead, it is based on our experimental results, namely, that merging the most confusable MDFA states yields dendrograms that really tell us important information about how the RNN hidden states are organized. We show two of these dendrograms in the paper (EMAILS and DATES). We emphasize strongly that these are not in any way cherry-picked examples. As stated in our response to Reviewer 1 above: “Our intention behind showing an EMAILS and DATES regular expressions that were formed outside of the aforementioned framework was to show how a typical, easily interpretable recognition algorithm is encoded by the RNN. We didn’t want the reader to be distracted by the regular expression itself but rather bring light to the interpretation of the dendrograms in section 4.5.” In order to alleviate any concerns, we will include figures of the randomly sampled MDFAs and corresponding dendrograms in the Appendix in the final version of the paper.
> > >
> > > As to recognition accuracy, 81% of the RNNs in the linear decoding experiments met the  minimum language recognition test accuracy of 95%. If we reduce the threshold to 90%, the fraction increase to 89%.

---

> > > > ### Comment · AnonReviewer3 · 2018-12-07
> > > > **concerns remain**
> > > >
> > > > To summarize my understanding of the author's rebuttal, they're saying that the key result isn't that linear decoders achieve high accuracy in decoding the abstract DFA states, but is instead that the abstract DFAs that are recovered from the "hierarchical clustering" process bear some kind of resemblance to the original DFA. Three points about this
> > > >
> > > > 1.) If this interpretability of the "clusters" is the real crux of the paper, instead of the decodeability referred to in the title, then the title and introduction of the paper really should reflect this.
> > > > 2.) I'm not sure what the integers and percentages inside the DFA state diagrams in figure 6 are (I asked about this in my original review but I didn't see an answer unfortunately). As a result, I don't know how the authors mean to interpret the dendrograms built on top of the state diagrams.
> > > > 3.) Without knowing exactly what interpretation the authors intend to draw from those dendrograms I don't want to be too categorical about this, but I will say that whatever the interpretation is, its seems very likely to be subject to the cherry-picking issue that R2 brought up. It seems to me like drawing any useful general conclusion from these two examples would be challenging.
> > > >
> > > > To summarize, (1) the authors responses to my and R2s questions/criticisms suggest that main text of the paper obscures the basic logic of the work, and (2) that basic logic seems to rest entirely on the interpretation of just two examples.
> > > >
> > > > Both of these points seem quite problematic, so at this time my score remains below the acceptance recommendation threshold.

---

> ### Author Response · Authors · 2018-11-26
> **Thank you for the in-depth questions and comments 2/2**
>
> -The regular expression in Figure 6 is incorrect.
> We thank the reviewer for finding this error. We will replace it with the correct regular expression “[a-d]+@[a-d]+.[v-z]{2,3}” in the final version.
>
> -How come Figure 3a goes up to 1.1? Isn’t it bounded by 1?
> You are correct that the decoding accuracy mean chart in Figure 3a is bounded by 1. The reason for the unbounded nature is that the error bars represent one standard deviation above and below the estimate of the mean accuracy, which doesn’t necessarily respect the bound as we modeled it as a Gaussian random variable. We agree with the reviewer that the top error bars should be bounded by 1 and will fix this in the final version by using a more appropriate representation such an interquartile ranges.
>
>
> -It is not clear how the shuffling of the characters is considered an independent distribution. The negative sampling procedure should appear in the main text.
> We believe the reviewer is referring to the term “independent” used in the Appendix under the “Data Generation” section, which is unclear. We did not intend to evoke the statistical meaning, but rather to explain how the two sampling procedures are different. In the camera-ready version of the paper we will replace the word in the sentence with “much different” to clarify.

---

### Official Review · AnonReviewer1 · 2018-11-04
**Interesting exploratory research, some more examples are desired**

**Rating:** 7
**Confidence:** 3

**Review:**

This paper investigates internal working of RNN, by mapping its hidden states
to the nodes of minimal DFAs that generated the training inputs and its
abstractions. Authors found that in fact such a mapping exists, and a linear
decoder suffices for the purpose.
Inspecting some of the minimal DFAs that correspond to regular expressions,
induced state abstractions are intuitive and interpretable from a viewpoint of
training RNNs by training sequences.

This paper is interesting, and the central idea of using formal languages to
generate feeding inputs is good (in fact, I am also doing a different research
that also leverages a formal grammar with RNN).

Most of the paper is clear, so I have only a few minor comments:

- In Figures 4 and 5, the most complex MDFA of 14 nodes does not have the
  lowest testing accuracies. In other words, testing accuracies is not
  generally proportional to the complexity of MDFA. Why does this happen?

- As noted in the footnote in page 5, state abstraction is driven by the idea
  of hierarchical grammars. Then, as briefly noted in the conclusion, why not
  using a simple CFG or PCFG to generate training sequences?
  In this case, state abstractions are clear by definition, and it is curious
  to see if RNN actually learns abstract states (such as NP and VP in natural
  language) through mapping from hidden states to abstracted states.

- Because this paper is exploratory, I would like to see more examples
  beyond only the two in Figure 6. Is it possible to generate a regular
  expression itself randomly to feed into RNN?

---

> ### Author Response · Authors · 2018-11-26
> **Thank you for your feedback**
>
> We appreciate the reviewers' comments and suggestions. If the reviewer has any additional follow-up comments or questions, we welcome them.
>
> -Why are the testing accuracies not generally proportional to the complexity of the MDFA? The most complex MDFA of 14 nodes does not have the lowest testing accuracies.
> In Figure 4, the testing accuracies are not proportional to the complexity of the MDFA due to our method of generating MDFAs in our experiments. Regular expressions are randomly generated by our pipeline and the resulting MDFA is created from the regular expression. We choose to sample in the space of regular expressions as opposed to the space of DFAs because sampling in regular expression space is more meaningful; that is, a valid regular expression that is generated is guaranteed to result in a DFA with desired behavior. If we were to sample in DFA space, it is possible that the resulting DFAs may have had unreachable states and other undesirable behavior. There is, however, no straightforward relationship in terms of complexity between MDFAs and their corresponding regular expressions, leading to the slight differences in proportionality seen in Figures 4 and 5.
>
> -Why not use a simple CFG or PCFG to generate training sequences?
> We choose regular expressions to generate training sequences for their simplicity, as they allow us to interpret the hidden state of the RNN in terms of the clearly defined states that constitute a regular expressions’ corresponding DFA. There is a substantial amount of literature on the relationship between RNNs and DFAs, but given the little literature surrounding complex regular expressions and DFAs, we want to rigorously explore this space before moving to grammars further up the Chomsky Hierarchy, such as CFGs. Using a CFG or PCFG is a logical next step for our work and is indeed a motivating example.
>
> -Is it possible to generate a regular expression randomly to feed into the RNN?
> Yes, is it possible to randomly generate the regular expressions. In our paper, we have developed a framework (Figure 1) for randomly generating regular expressions. At the bottom of section 4.1, we mention that the experiments and results we present are utilizing a dataset of ~500 randomly generated regular expressions in order to get the statistically significant results required in section 4.2, 4.3, and 4.4.
>
> -It would be nice to provide more examples?
> We agree with this suggestion. Due to space constraints, we did not include more in the main text. We will add more examples to the appendix of the final version of the paper.

---

### Public Comment · (anonymous) · 2018-10-29
**Nice paper**

This is a nice piece of work, well-written, on a hot topic, providing an interesting novel approach and some important insights.

I would like to point out 2 recent works on the matter that could be interesting to discuss in the paper if accepted:

- In [1], the authors prove the equivalence between linear 2-order RNN and weighted automata. The linearity restriction clearly echoes the one of this paper.

- In [2], the authors show that non-linear RNN can be efficiently approximated by weighted automata, suggesting as strong link between the states of the automata and the inner representation of RNN, as in this paper.

[1] Connecting Weighted Automata and Recurrent Neural Networks through Spectral Learning, Guillaume Rabusseau, Tianyu Li, Doina Precup, https://arxiv.org/abs/1807.01406

[2] Explaining Black Boxes on Sequential Data using Weighted Automata, Stephane Ayache, Remi Eyraud, Noe Goudian, https://arxiv.org/abs/1810.05741

---

> ### Author Response · Authors · 2018-12-05
> **Thank you**
>
> Thank you for pointing us in the direction of these recent works. Both of them encompass the general connection between RNNs and Automata in some way which we believe is a fruitful area of research relating to interpretable models. One extension of our work would be to utilize other RNN frameworks, such as Giles 2nd order RNN which we would hypothesis would likely encode an automata with more accuracy than vanilla RNNs because of it's original intention to do so. We believe that our work can be thought of a parallel to the work in [1]. The work in [2] seems equally important and relevant. Although our work is not focused on extracting Automata from RNNs but rather relating the underlying representations, we believe our work resonates with this paper as well. We will cite both of these papers in our related works section.

---

### Meta-Review · Area_Chair1 · 2018-12-13
**Acceptable**

**Confidence:** 4
**Recommendation:** Accept (Poster)

**Metareview:**

This paper presents experiments showing that a linear mapping existing between the hidden states of RNNs trained to recognise (rather than model) formal languages, in the hope of at least partially elucidating the sort of representations this class of network architectures learns. This is important and timely work, fitting into a research programme begun by CL Giles in 92.

Despite its relatively low overall score, I am concurring with the assessment made by reviewer 1, whose expertise in the topic I am aware of and respect. But more importantly, I feel the review process has failed the authors here: reviewers 2 and 3 had as chief concern that there were issues with the clarity of some aspects of the paper. The authors made a substantial and bona fide attempt in their response to address the points of concern raised by these reviewers. This is precisely what the discussion period of ICLR is for, and one would expect that clarity issues can be successfully remedied during this period. I am disappointed to have seen little timely engagement from these reviewers, or willingness to explain why they are stick by their assessment if not revisiting it. As far as I am concerned, the authors have done an appropriate job of addressing these concerns, and given reviewer 1's support for the paper, I am happy to add mine as well.